# Influence of the Coupling South Atlantic Convergence Zone-El Niño-Southern Oscillation (SACZ-ENSO) on the Projected Precipitation Changes over the Central Andes

**Juan C. Sulca** [1,*] and **Rosmeri P. da Rocha** [2]

1   Subdirección de Ciencias de la Atmósfera e Hidrósfera, Instituto Geofísico del Perú, Lima 15026, Peru
2   Department of Atmospheric Sciences, University of São Paulo, São Paulo 05508-090, Brazil; rosmerir.rocha@iag.usp.br
*   Correspondence: jsulca@igp.gob.pe

**Abstract:** There are no studies related to the influence of the coupling between the South Atlantic Convergence Zone (SACZ) and El Niño-Southern Oscillation (ENSO) pattern variability on future changes in the austral summer (December-February, DJF) precipitation over the central Andes. Therefore, we evaluated the historical simulations (1980–2005) and projections (2070–2099) for the Representative Concentration Pathway 8.5 (RCP 8.5) scenario of 25 global climate models (GCMs) from the Coupled Model Intercomparison Project phase 5 (CMIP5). Moreover, we also consider the Regional Climate Model version 4 (RegCM4) projections nested in three CMIP5 GCMs (GFDL-ESM2M, MPI-ESM-MR, and HadGEM2-ES) under RCP 8.5. We separate the CMIP5 GCMs according to their abilities to simulate the nonlinear characteristics of ENSO and the SACZ for the historical period. We found that only three out of 25 CMIP5 GCMs (hereafter group A) simulate the nonlinear characteristics of ENSO and the SACZ during the historical period. Although most CMIP5 GCM project DJF precipitation decreases over the central Andes, group A project precipitation increases related to the projected increase in deep convection over the central Peruvian Amazon. On the regional scale, only RegGFDL (nested in a group A CMIP5 GCM) projects a statistically significant increase in DJF precipitation (~5–15%) over the northern central Andes and the central Peruvian Amazon. Conversely, all RegCM4 simulations project a decrease in DJF precipitation (~−10%) over the southern central Andes.

**Keywords:** austral summer precipitation; Central Andes; climate change

## 1. Introduction

The South American Monsoon System (SAMS) is one of the major monsoon systems in the Southern Hemisphere [1]. The SAMS onset occurs in October, reaching its mature phase from December to February and declining in April [2]. During the austral summer (December-January-February, DJF), the maximum precipitation falls over the south-central Amazon (Figure 1a). Simultaneously, there is an intense low-level moisture flux from the central Amazon basin toward southeastern South America at 850 hPa called the South American low-level jet (SALLJ; [3]) (Figure 1a). In the upper troposphere (200 hPa), the Bolivian high-Nordeste low (BH-NL) system is the main feature of the austral summer climatology in South America [4] (Figure 1b). During the austral winter, the BH-NL system disappears when the upper-level westerly zonal flow prevails over the continent.

Precipitation over the South American continent features a northwest-southeastward band of convective activity from the Amazon basin to the western South Atlantic (20°–40° S, 50°–20° W), called the South Atlantic convergence zone (SACZ). The SACZ is present all year; however, its maximum intensity occurs during the austral summer [5–7] (Figure 1a). The SACZ modulates the intensity and forms the basic shape of the BH by releasing condensational heat to the upper troposphere [8]. SACZ also is associated with

dry anomalies over the northern central Andes and the equatorial Amazon basin during the austral summer [9].

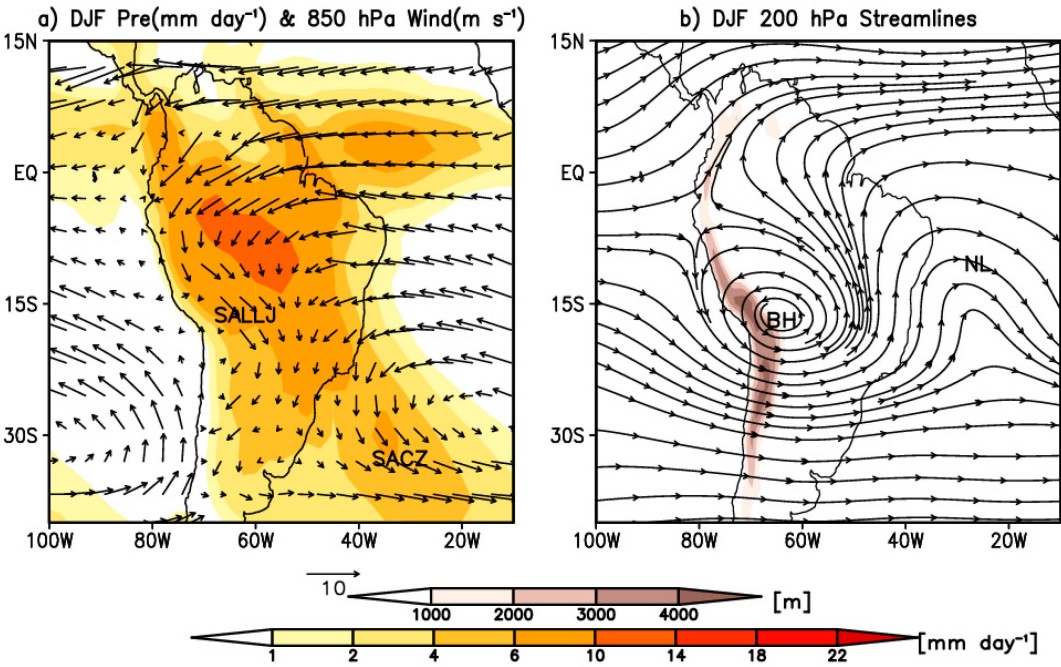

**Figure 1.** Austral summer climatology (1981–2005) of (**a**) 850 hPa wind (m s$^{-1}$) and precipitation (mm day$^{-1}$) and (**b**) 200 hPa streamlines as provided by NCEP/NCAR reanalysis and GPCP datasets. SALLJ means South American low-level jet. SACZ means South Atlantic convergence zone. BH means Bolivian high. NL means Nordeste Low. The 1000- and 4000-m topographic contours are indicated by brown shading. This Figure is an adaptation from Sulca et al. (2021) with permission of the American Meteorological Society.

The central Andes run continuously along the western coast of South America from 18.5° S to 8° S. The mean maximum (peak) height of the central Andes exceeds 5000 m MSL. The precipitation in the central Andes has a wet period between September and April of the following year. The rainy season in the central Andes occurs during DJF, while the dry season occurs between June and August [10,11]. The summer precipitation in the central Andes is associated with the enhanced easterly flow at upper levels, while an anomalous westerly flow over the Altiplano controls the dry season [12–14].

Although the El Niño-Southern Oscillation (ENSO) impacts temperature and precipitation around the world, few studies have investigated the impacts of diverse types of ENSO events on precipitation in Peru [15,16]. The positive central El Niños inhibit precipitation over the central Andes and the equatorial Amazon east of 70° W. Concerning positive Eastern El Niño episodes, the warm conditions over the far-eastern Pacific induce precipitation over the coasts of southern Ecuador and northern Peru while inhibiting precipitation over the Peruvian Andes south of 8° S. However, the reduction of precipitation over the Peruvian Andes is only statistically significant over the Peruvian Altiplano. Conversely, both Central and Eastern La Niñas induce a precipitation increase over the central Andes [15]. Moreover, ENSO is an essential modulator of the upper-level zonal wind over Peru during the austral summer [9].

Several studies have evidenced that the Bjerknes feedback (e.g., westerly anomalies lead to warming surfaces, which reinforce westerly anomalies through an anomalous Walker Pacific cell [17]) is a nonlinear process in the central and eastern tropical Pacific [18,19]. The nonlinear Bjerknes feedback is needed for moderate El Niño episodes to become strong El Niño episodes once the temperature (Ts) exceeds a critical temperature (Tc) called Ccrit and Ecrit in the central and eastern tropical Pacific, respectively [18]. The

Ecrit (Ts $\geq$ 26 °C) is the temperature associated with the activation of large-scale deep convection over the far-eastern Pacific, which is climatologically cold [20].

Climate change will affect the temporal and spatial patterns of rainfall in South America, with significant consequences for water resources in the tropical Andes [21]. Since a high percentage of annual rainfall can fall in a few days, causing soil erosion, landslides, or floods, a tiny portion of rainfall over a considerable period may impact agriculture [21,22]. The future increase in evapotranspiration will produce changes in the regional precipitation pattern and the frequency of extreme hydroclimatic events such as floods or droughts [23]. These same changes also produce variations in several hydrological parameters (e.g., streamflow, soil moisture, and groundwater storage) and the availability of potable water for the population [24].

Global climate models (GCMs) are the main tools for simulating climate changes and estimating future conditions on global and regional scales. Several studies have evaluated the Coupled Model Intercomparison Project phase 5 (CMIP5; [25]) GCMs in simulating different climatological characteristics of the equatorial Pacific Ocean and South America [26–29].

For the equatorial Pacific Ocean, few CMIP5 GCMs simulate the ENSO nonlinearity and the nonlinear behavior of the Central and Eastern El Niños [29]. The CMIP5 GCMs that simultaneously simulate the nonlinear Bjerknes feedback in the central and eastern Pacific show that the nonlinear Bjerknes feedback is needed to simulate the observed change of the Pacific Walker cell structure [29].

The results of the analysis of the nonlinear behavior of ENSO and how it will impact the rainfall of the equatorial Pacific in the far future using pre-industrial control simulations from several CMIP5 GCMs pointed out that the nonlinearity of ENSO can be quantified through the parameter "$\alpha$" [30]. The parameter $\alpha$ is the leading coefficient of the quadratic fit of the form $PC2 = \alpha * PC1^2 + b * PC1 + c$ in the PC1-PC2 plane. The authors found that general circulation models with $\alpha$ greater than -0.12 ($\alpha_{obs}$) simulate ENSO feedback in close agreement with the observations. Besides, AMIP and historical simulations from the CMIP5 models pointed out that the nonlinearity of ENSO is associated with the intensity of the mean-state of the Pacific Walker Cell [29].

In terms of South America, the Intergovernmental Panel on Climate Change [31], who performed a multimodel simulation with all CMIP5 GCMs, found that slight increases in DJF precipitation were projected over the central Andes and the central Peruvian Amazon, but that the increase was statistically significant only over the central Peruvian Amazon. However, they did not assess the validation of each CMIP5 GCM for precipitation over the central Andes.

New studies about the performance of CMIP5 GCMs in the simulation of several climatological characteristics of South America have appeared after the publication of the IPCC report in 2013. For instance, five CMIP5 GCMs [BCC-CSM1(m), CCSM4, CESM1-CAM5, CNRM-CM5, and EC-EARTH] reproduce the SALLJ during the austral summer [28]. Most CMIP5 GCMs simulate the BH over South America during the austral summer, except the CNRM-CM5 model, which simulates the BH southeastward of its climatological position [28]. Some CMIP5 GCMs have problems simulating the climatological position and orientation of the SACZ during the austral summer [27]. These same CMIP5 GCMs also have problems in the simulation of the Atlantic Intertropical Convergence Zone (ITCZ) caused by the simulation of the SACZ and other model problems.

Moreover, several studies evaluated the CMIP5 GCMs in the simulation of precipitation over South America [10,32–34]. Most CMIP5 GCMs underestimate large-scale precipitation during the wet season [33]. According to weighting score analysis, three CMIP5 GCMs (INMCM4, MIROC5, and IPSL-CM5A-LR) reproduce austral summer precipitation over the Bolivian Andes but present a strong wet bias [32]. Besides, three CMIP5 GCMs (MPI-ESM-LR, MIROC5, and CCSM4) successfully reproduce the large-scale atmospheric circulation (BH) over South America and, more specifically, over Bolivia [35]. Conversely, two CMIP5 GCMs (IPSL-CM5A-LR and HadGEM2-ES) do not reproduce the

characteristics of the first three CMIP5 GCMs mentioned above. Since the BH is not enough to describe the pattern of atmospheric circulation associated with DJF precipitation over the northern central Andes (e.g., Mantaro basin) [36], we focus on the performance of the BH-NL system over South America during the austral summer in CMIP5 GCMs.

The climate will change as a result of its complex interaction with ecosystems and human activities. To characterize a range of plausible climate futures, the scientific community develops and uses future scenarios such as the representative concentration pathways (RCPs, [37]). In terms of modeling, each RCP scenario has a prescribed increase in energy in the earth's climate system by 2100 relative to preindustrial levels [38]. There are four RCPs (2.6, 4.5, 6.0, and 8.5). RCP 8.5 means that the greenhouse gas emissions and concentrations increase considerably over time, leading to a radiative forcing of +8.5 W m-2 at the end of the 21st century. The RCP 8.5 scenario also is known as "baseline" scenario because that does not include any specific climate mitigation target [39].

Since CMIP5 GCMs have a coarse horizontal resolution (~100–300 km) for estimating the future change in precipitation over complex topography, such as the central Andes, a dynamical downscaling (DD) technique was applied by running regional climate models (RCMs) at a higher resolution driven by GCMs as their lateral boundaries. Using the HadCM3 model as boundary conditions, the Eta model projects a gradual increase in water deficit and precipitation concentration in the northern part of the central Andes (e.g., the Mantaro basin) under the A1B scenario [40]. Conversely, several single-RCM dynamical downscaling studies found, on average, a reduction in precipitation over the southern central Andes for the end of the 21st century [41].

On the other hand, the statistical downscaling techniques are formed by two main groups: empirical-statistical downscaling [42] and bias correction [43]. The empirical-statistical downscaling (ESD) techniques use the linear relationship between local variables and large-scale forcings estimated by GCMs variables in the present period to calculate future precipitation changes [42]. Using the linear relationship between precipitation (PRE) and 200 hPa zonal wind (U200) anomalies in the present climate, the austral summer precipitation over the central Andes projects a significant reduction between 10% and 30% by the end of this century under the A2 scenario [11]. However, the results of the ESD technique must be taken with caution because the PRE-U200 relationship over the central Andes started weakening in 2000 [44].

Bias correction techniques allow us to estimate the impact of climate change on water resources at a basin-scale [43]. Bias-correction methods determine the difference in the climate outputs (mean, variability) between the current and future conditions from outputs of global climatic models (GCMs) [45]. However, the outcomes of the bias correction techniques must be interpreted carefully when the region presents local gradients as in the case of the Tropical Andes [21]. Besides, most of the bias correction studies use the Taylor's diagram as a performance criterion for the final selection of the GCMs analyzed. Based on the multimodel ensemble of five (EC-EARTH, HadGEM2-ES, IPSL-CM5A-LR, MIROC5, and MPI-ESM-LR) out of 31 CMIP5 GCMs, the Lake Titicaca basin projects an increase in austral summer precipitation below 8.2% for the 2034–2064 period relative to 1984–2014 under the RCP 8.5 scenario [46].

The RegCM3 simulations for the South American domain project a reduction in summer precipitation in the Altiplano region for the far future (2070–2100) under the RCP 8.5 scenario [47]. The same RegCM3 simulations, however, do not project the same sign of summer precipitation over the northern part of the central Andes and the central Peruvian Amazon [47]. The multimodel ensemble of RCM simulations from the Coordinated Regional Climate Downscaling Experiment (CORDEX) projects a reduction of DJF precipitation along the Peruvian Andes and in the central Peruvian Amazon around of −2 mm day-1 under the RCP 8.5 scenario [48]. According to several RegCM4 simulations, three out of four clusters of the DJF precipitation of the Altiplano region project a negative trend [49]. However, the authors did not assess the influence of ENSO and SACZ on the projections of the DJF precipitation along the tropical Andes. Moreover, these studies did not assess the

large-scale mechanism associated with the contrast of signs of the projected precipitation change between southern and northern parts of the central Andes.

Motivated by these considerations, this study aims to identify CMIP5 GCMs that simulate the nonlinear ENSO characteristics and the SACZ to quantify the future change (2070–2099 compared to 1980–200 period) in austral summer precipitation over the central Andes. This study's main goal is to help as input for elaborating new water management plans in the central Andes of Peru, which is a region that is very vulnerable to climate change.

This paper is structured as follows. Section 1 contains the scientific background for our study. Section 2 describes the data and methods that were applied. Section 3 describes (a) the validation of the SACZ in CMIP5 GCMs, (b) projected changes in precipitation and low- and upper-level winds over the equatorial Pacific Ocean and South America, and (c) projected rainfall changes in the central Andes using RegCM4 simulations. Section 4 discusses the new findings of this study. The final section summarizes the significant findings.

## 2. Materials and Methods

### 2.1. Data

This study used global and regional monthly gridded precipitation datasets. For South America, we used monthly gridded precipitation from the Global Precipitation Climatology Project (GPCP) described in [50]. The GPCP dataset has a horizontal resolution of $2.5° \times 2.5°$ and covers the 1974–2020 period.

We used zonal and meridional winds at 850 and 200 hPa from the National Centers for Environmental Prediction (NCEP)-National Center for Atmospheric Research (NCAR) reanalysis data [51] with $2.5° \times 2.5°$ grid spacing for the 1980–2005 period.

The convection index associated with the SACZ was based on monthly gridded outgoing longwave radiation (OLR) data from the NCEP/National Oceanic and Atmospheric Administration (NOAA) [52]. The OLR data have a spatial resolution of $2.5° \times 2.5°$ and cover the 1974–2020 period.

The topography dataset of 1 km of resolution from the Shuttle Radar Topography Mission (SRTM, [53]) was rescaled to 50 km to be compared with the topography of 50 km of the RegCM4 model.

Daily rainfall data of 0.25° resolution called Climate Hazards Group Infra-Red Precipitation with Station data (CHIRPS, [54]) is rescaled to 50 km to be compared with RegCM4 experiments. CHIRPS is based on the combination of TRMM (TMPA 3B42 v7) and station-based observations and also covers the 1980-2005 period.

### 2.2. CMIP5 GCMs

This study also used monthly atmospheric variables [precipitation, zonal and meridional winds at 850 and 200 hPa, and top of the atmosphere (TOA)-outgoing longwave flux] from historical simulations and Representative Concentration Pathway 8.5 (RCP 8.5) projections from 25 CMIP5 GCMs [55]. The CMIP5 simulations were downloaded from the Centre for Environmental Data Analysis (CEDA). The climatology period is 1980–2005, and the far-future period is 2070–2099. The details of the 25 CMIP5 GCMs are shown in Table 1.

**Table 1.** CMIP5 GCMs used in this study: modeling center, official name, mean horizontal resolution of the atmospheric component, and available data types.

| Modeling Center | Model Name | Atmosphere Resolution (°) | Historical | RCP 8.5 |
|---|---|---|---|---|
| CSIRO and Bureau of Meteorology (BOM), Australia | ACCESS1-0 | 1.875° × 1.25° | X | X |
| | ACCESS1-3 | 1.875° × 1.25° | X | X |
| College of Global Change and Earth System Science, Beijing Normal University | BNU-ESM | 2.8125° × 2.7906° | X | X |
| NOAA/Geophysical Fluid Dynamics Laboratory | GFDL-CM3 | 2.5° × 2° | X | X |
| | GFDL-ESM2M | 2.5° × 2.0225° | X | X |
| NASA Goddard Institute for Space Studies | GISS-E2-R | 2.5° × 2° | X | X |
| | GISS-E2-H | 2.5° × 2° | X | X |
| | GISS-E2-R-CC | 2.5° × 2° | X | X |
| National Center for Atmospheric Research | CCSM4 | 1.25° × 0.9424° | X | X |
| Centre National de Recherches Meteorologiques/Centre Europeen de Recherche et Formation Avancee en Calcul Scientifique | CNRM-CM5 | 1.40625° × 1.4008° | X | X |
| | CNRM-CM5-2 | 1.40625° × 1.4008° | X | X |
| Centro Euro-Mediterraneo per I Cambiamenti Climatici Model CMS | CMCC-CESM | 3.75° × 3.4431° | X | X |
| | CMCC-CMS | 3.75° × 3.7111° | X | X |
| Institute of Atmospheric Physics (IAP) of the Russian Academy of Sciences | INM-CM4.0 | 2° × 1.5° | X | X |
| Institute for Numerical Mathematics L'Institut Pierre-Simon Laplace | IPSL-CM5A-LR | 3.75° × 1.8947° | X | X |
| | IPSL-CM5A-MR | 2.5° × 1.2676° | X | X |
| | IPSL-CM5B-LR | 3.75° × 1.8947° | X | X |
| Environmental Studies, and Japan Agency for Marine-Earth Science and Technology Meteorological Research Institute | MIROC5 | 1.40625° × 1.4008° | X | X |
| Max Planck Institute | MPI-ESM-LR | 1.875 × 1.8653 | X | X |
| | MPI-ESM-MR | 1.875 × 1.8653 | X | X |
| | MPI-ESM-P | 1.875 × 1.8653 | X | X |
| Research Council of Norway | NorESM1-M | 2.5° × 1.8947° | X | X |
| Hadley Centre from the United Kingdom | HadCM3 | 2.5 × 3.75 | X | - |
| | HadGEM2-ES | 1.25 × 1.875 | X | X |
| Beijing Climate Center (BCC), Chinese Meteorological Administration (CMA), China | BCC-CSM1(m) | 2.8125 × 2.7906 | X | X |
| European Community Earth-System Model, Europe | EC-EARTH | 1.125 × 1.1215 | X | X |
| National Science Foundation (NSF)–U.S. Department of Energy (DOE)–NCAR, United States | CESM1-CAM5 | 1.25 × 0.9424 | X | X |

*2.3. RegCM4 Simulation*

The domain of South America in the CORDEX-RegCM4 simulations covers the region 96–22.5° W, 59.5° S–11° N [56,57]. The RegCM4 simulations have a horizontal resolution of approximately 50 km × 50 km (192 × 202 grid points in east−west and north−south directions, respectively) and 18 sigma-pressure levels. These simulations are part of the CREMA (CORDEX REgCM4 hyper-MAtrix) experiment described in detail by [58]. The CREMA objective was to provide a mini-ensemble of RegCM4 experiments using different driving CMIP5 GCMs, different RegCM4 physics configurations, and different

scenarios. In the initial phase of CREMA, three CMIP5 GCMs were selected due to the availability of the 6-hourly fields needed to run RegCM4. The three GCMs are HadGEM2-ES (hereafter HadGEM2; [59]), MPI-ESM-MR (hereafter, MPI; [60]), and GFDL-ESM2M (hereafter GFDL; [61]). The RegCM4 simulations are labeled RegGFDL, RegMPI, and RegHadGEM2.

The RegCM4 model used the greenhouse gas concentration of RCP 8.5, the Community Land Model version CLM3.5 [62], and the Emanuel and Zivkovic-Rothman convective scheme (CLM-Emanuel, [63]). This study analyzed two periods: climatology (1980–2005) and far future (2070–2099) though the RegHadGEM2 simulation had a far-future period of 2070–2098.

*2.4. Methods*

We verified the nonlinear characteristics of ENSO in the three CMIP5 GCMs (MPI-ESM-MR, MPI-ESM-P, and HadGEM2-ES) following the methodology proposed in [29]. First, the nonlinear ENSO metric ($\alpha$) is equal to $-0.12$. Second, the nonlinear behavior of both ENSO Pacific sea surface temperatures (SSTs) (Central and Eastern) is verified. These three CMIP5 GCMs expand in the list of CMIP5 GCMs analyzed in [29].

The observed SACZ pattern is calculated using the monthly gridded OLR data from NOAA/NCEP [38]. We use the first principal component (PC) of DJF OLR anomalies over central-eastern Brazil region delimited by 65–30° W, 40° S–0° N [9,64]. To quantify the performance of the CMIP5 GCMS in the simulation of the SACZ pattern, we computed the spatial correlation between the observed and simulated SACZ patterns.

The performance of the CMIP5 GCM projections of DJF precipitation change over the entire central Andes is evaluated by defining three different groups of CMIP5 GCMs based on their abilities to represent the nonlinear ENSO characteristics and the SACZ. Group A contains the CMIP5 GCMs that simulate the nonlinear ENSO characteristics and the SACZ, while group B contains CMIP5 GCMs that simulate the nonlinear ENSO characteristics but not the SACZ. The remaining CMIP5 GCMs are in group C (Table 2).

We discuss the composite analysis of the absolute difference between future (2070–2099) and present (1980–2005) climates of DJF precipitation and circulation over the entire central Andes [65]. To assess the statistical significance of the differences in DJF precipitation, we computed the two-tailed Student's t-test for the difference of means at a 0.1 significance level [66].

**Table 2.** List of CMIP5 GCM groups used in this study.

| CMIP5 GCMs | | |
|---|---|---|
| Group A (Nonlinear ENSO Characteristics + SACZ) | Group B (Nonlinear ENSO Characteristics) | Group C |
| BNU-ESM, CCSM4, GFDL-ESM2M | CMCC-CMS, CMCC-CESM, CNRM-CM5, GISS-E2-R, GFDL-CM3 | ACCESS1-0, ACCESS1-3, CanESM2, FIO, GISS-E2-H-CC, GISS-E2-H, INMCM4, IPSL-CM5A-MR, IPSL-CM5A-MR, IPSL-CM5B-LR, MIROC5, MPI-ESM-P, MPI-ESM-LR, MPI-ESM-MR, NorESM1-M, HadCM3, HadGEM2-ES |

## 3. Results

*3.1. Performance of the CMIP5 Models in Simulating the SACZ*

Before this analysis, we found that three CMIP5 GCMs (MPI-ESM-MR, MPI-ESM-P, and HadGEM2-ES) do not reproduce the nonlinear behavior of ENSO because they present a nonlinear ENSO metric ($\alpha$) that is greater than $-0.12$ or even positive (0.03, $-0.01$, and $-0.02$, respectively). These three CMIP5 models were included in the remaining analysis. These results show that eight out of 25 CMIP5 GCMs reproduce nonlinear

ENSO characteristics. This result implies that the three CMIP5 GCMs reproduce the observed mean state of the Pacific Walker cell and the observed atmospheric teleconnections associated with austral summer precipitation over the central Andes with ENSO (e.g., change of the upper-level zonal flow [12,13,67]).

The DJF climatology of precipitation characterizes the SACZ as the northwest-southeast band of more intense rainfall from the western Amazon to the southwest South Atlantic Ocean (Figure 1a). Low-level circulations show northeasterly winds over the Amazon turning northwesterly near the Andes Mountains, following this direction until southeastern South America. At the upper levels, the establishment of the BH followed downstream by the northeast Brazil low is noted (Figure 1b).

Figure 2a presents the OLR pattern (first EOF) of the SACZ for the NCEP/NOAA reanalysis. The SACZ features positive OLR anomalies over the equatorial and central sectors of the Amazon basin while negative OLR anomalies predominate over the coast of northeastern Brazil southward of $12°$ S. The positive OLR anomalies indicate that SACZ suppresses convection over the Amazon basin, verifying the regional signal of the SACZ documented in previous numerical studies [7,68].

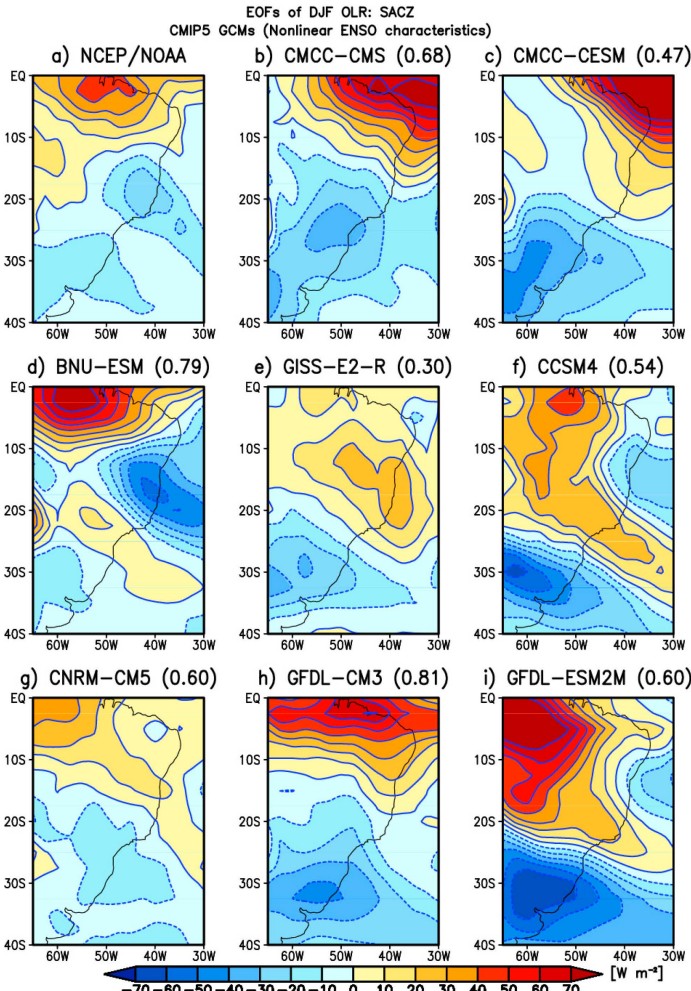

**Figure 2.** First leading EOF of DJF (1980–2005) OLR (W m$^{-2}$) anomalies inside the region (65°–30° W, 40° S–0° N) for (**a**) NCEP/NOAA, (**b**) CMCC-CMS, (**c**) CMCC-CESM, (**d**) BNU-ESM, (**e**) GISS-E2-R, (**f**) CCSM4, (**g**) CNRM-CM5, (**h**) GFDL-CM3.1, and (**i**) GFDL-ESM2M models. The numbers within the parenthesis represent the spatial correlation of the DJF OLR pattern between the observed and each individual CMIP5 GCM. This analysis is based on historical simulations from eight CMIP5 GCMs that simulate nonlinear ENSO characteristics.

We also present the first EOF of OLR for the 8 CMIP5 GCMs that simulate the nonlinear characteristics of ENSO (Figure 2) and 17 CMIP5 GCMs that cannot simulate nonlinear ENSO characteristics (Figure 3).

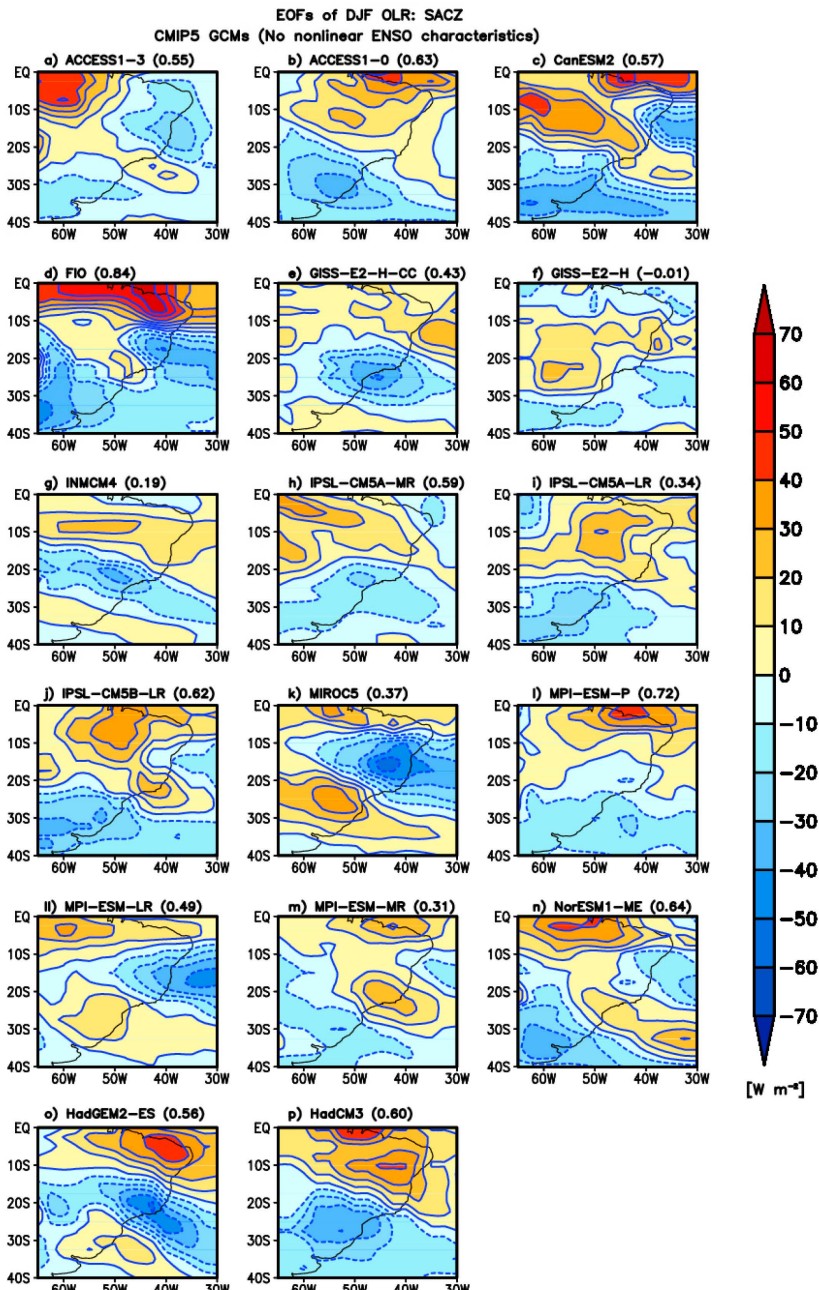

**Figure 3.** First leading EOF of DJF (1980–2005) OLR (W m$^{-2}$) anomalies inside the region (65°–30° W, 40° S–0° N) for (**a**) ACCESS1-3, (**b**) ACCESS1-0, (**c**) CanESM2, (**d**) FIO, (**e**) GISS-E2-H-CC, (**f**) GISS-E2-H, (**g**) INMCM4, (**h**) IPSL-CM5A-MR, (**i**) IPSL-CM5A-LR, (**j**) IPSL-CM5B-LR, (**k**) MIROC5, (**l**) MPI-ESM-P, (**ll**) MPI-ESM-LR, (**m**) MPI-ESM-MR, (**n**) NorESM1-ME, (**o**) HadGEM2-ES, and (**p**) HadCM3. The numbers within parenthesis represent the coefficients of the spatial correlation the DJF OLR pattern between the observed and each CMIP5 GCM. This analysis is based on historical simulations from 17 CMIP5 GCMs that cannot simulate nonlinear ENSO characteristics.

According to Figure 2, most CMIP5 GCMs that simulate nonlinear ENSO characteristics have a positive spatial correlation with the observed SACZ pattern above 0.50, except CMCC-CESM and GISS-E2-R. Although there is a high spatial correlation between observations and CMIP5 GCMs, only four of them (BNU-ESM, CCSM4, CNRM-CM5, and GFDL-ESM2M) reproduce the observed OLR pattern associated with the SACZ (Figure 2d,f,g,i). The remaining CMIP5 GCMs simulate an anomalous OLR dipole with negative anomalies over southeastern Brazil and positive anomalies over the Amazon basin and the equatorial Atlantic Ocean. In some of these models, the maximum positive signal of OLR prevails over the northern coast of northeastern Brazil and the equatorial Atlantic Ocean (Figure 2b,c,h), showing the lack of coupling of the SACZ with the Amazon basin and thus with the local Walker cell. These results highlight that only four out of 25 CMIP5 models reproduce the atmospheric teleconnections between the Amazon basin and SACZ during the austral summer.

Similar results are found for the CMIP5 GCMs that cannot simulate the nonlinear ENSO characteristics (Figure 3). Figure 3 shows that 11 (ACCESS1-3, ACCESS1-0, CanESM2, FIO, INMCM4, IPSL-CM5A-MR, IPSL-CM5B-LR, MPI-ESM-P, NorESM1-ME, HadGEM2-ES, and HadCM3) out of 17 CMIP5 GCMs were positively correlated (above 0.50) with the observed SACZ pattern. Nevertheless, only four of them (ACCESS1-3, FIO, IPSL-CM5A-MR, and NorESM1-ME) simulate positive OLR anomalies over the equatorial Amazon basin and central Amazon basin. Therefore, 17 out of 25 CMIP5 GCMs cannot simulate the EOF SACZ pattern, showing that the simultaneous simulation of the SACZ and ENSO is a great deficit in some CMIP5 GCMs.

To assess the impacts of the deficiency in the simulation of the SACZ on the simulation of regional atmospheric circulation over South America in CMIP5 GCMs that simulate the nonlinear characteristics of ENSO, Figure 4 presents a composite analysis of the DJF precipitation and 200 hPa streamlines for the 1980–2005 period. For precipitation, all CMIP5 GCMs overestimate the intensity of precipitation along the tropical Andes (Figure 4a–h), which might be associated with underestimating the Andes Cordillera height and parameterization deficiency in steeper topography [69]. Figure 4a–h also show that all CMIP5 GCMs that simulate nonlinear ENSO characteristics overestimate the precipitation over northeastern Brazil. Nevertheless, two CMIP5 GCMs (CMCC-CMS and GFDL-CM3) only reproduce the core of precipitation over the central Amazon basin, as in the GPCP data (Figure 4a,g). Several CMIP5 GCMs overestimate the DJF precipitation over the equatorial Atlantic Ocean off the coast of northernmost Brazil (Figure 4a,b,d,g), which might be related to a deficiency in the simulation of the upper-level Nordeste Low. Concerning the upper-level circulation, five out of eight CMIP5 GCMs (CMCC-CMS, BNU-ESM, CCSM4, GFDL-CM3, and GFDL-ESM2M) simulate both BH and NL systems (Figure 4b,d,f,h,i). Conversely, the CNRM-CM5 model simulates the NL southward of its observed position (Figure 4f), and the CMCC-CESM and GISS-E2-R models cannot represent the horizontal extension of the BH (Figure 4b,d). These results show that the simulation of the upper-level BH-NL system does not guarantee the simulation of the SACZ.

As discussed in the methodology, we show in Table 2 the three groups of CMIP5 GCMs: Group A contains three members (CMIP5 GCMs that simulate both the nonlinear ENSO characteristics and the SACZ), Group B contains five members (CMIP5 GCMs that simulate nonlinear ENSO characteristics but not the SACZ), and Group C contains 19 members (remaining CMIP5 GCMs).

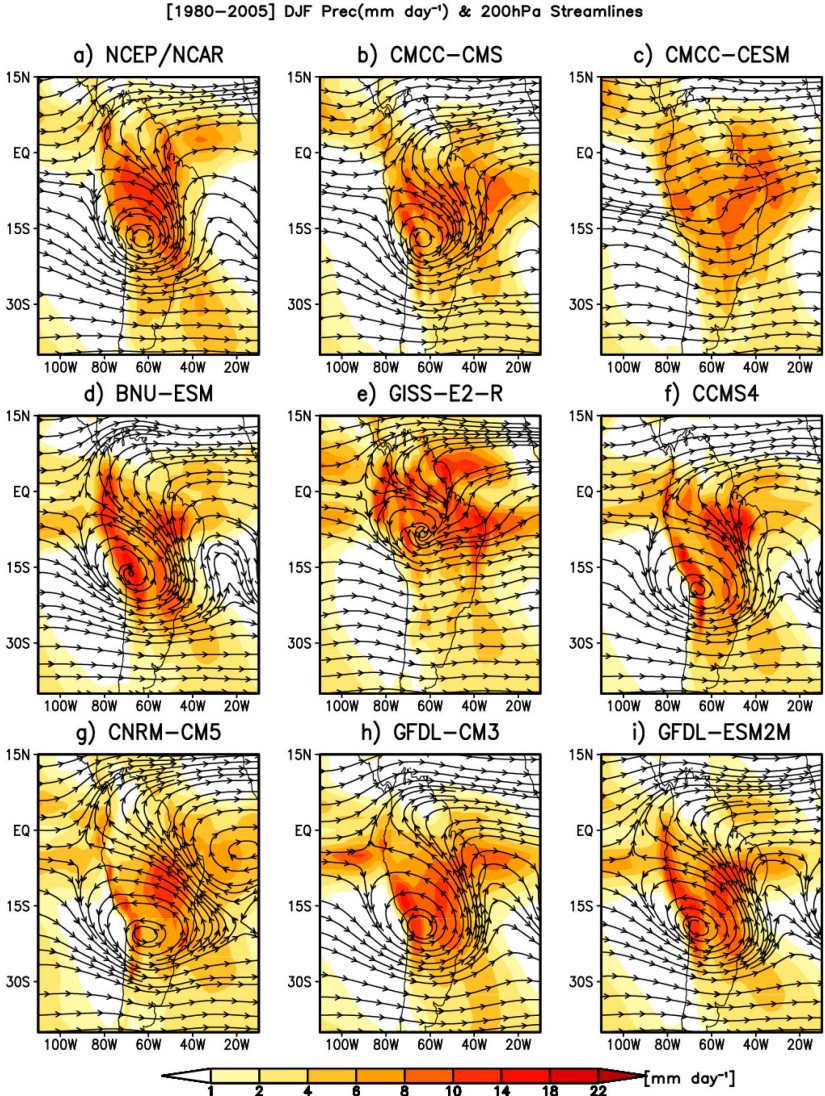

**Figure 4.** Climatology of DJF precipitation (mm day$^{-1}$) and 200 hPa streamlines in the current climate (1980–2005) for (**a**) NCEP/NCAR, (**b**) CMCC-CMS, (**c**) CMCC-CESM, (**d**) BNU-ESM, (**e**) GISS-E2-R, (**f**) CCSM4, (**g**) CNRM-CM5, (**h**) GFDL-CM3, and (**i**) GFDL-ESM2M models. This analysis is based on historical simulations from eight CMIP5 GCMs that simulate nonlinear ENSO characteristics. Analysis based in the 1980–2005 period.

### 3.2. Performance of the RegCM4 Model

When comparing the topography of the DEM data with the RegCM4, the topography of the RegCM4 underestimates the altitude of the entire Andes Mountains (Figure 5b). The topography of the RegCM4 does not present the DEM altitude above 4000 m along the central and northern parts of the central Andes northward of 14.2° S (Figure 5d). According to Figure 5d this also occurs in the northeastern Bolivian Altiplano. However, the topography of the RegCM4 is better than the original topography of the CMIP5 GCMs, which has elevation lower than 2500 m over the central and northern parts of the central Andes (Figures not shown).

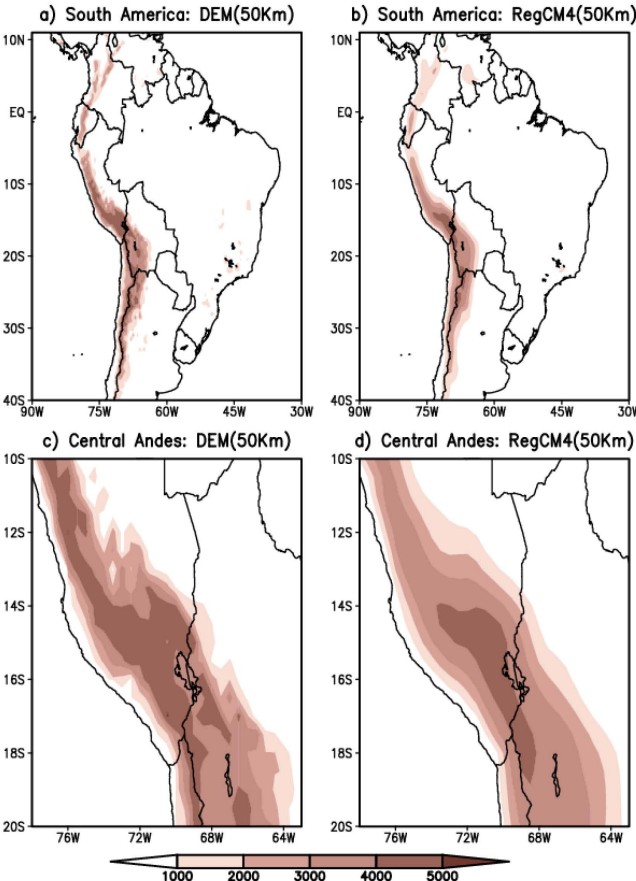

**Figure 5.** Topography of (**a**,**b**) South America and (**c**,**d**) central Andes for (**a**,**c**) DEM data, and (**b**,**d**) RegCM4 model. The 1000- and 5000-m topographic contours are indicated by brown shading.

Concerning summer, CHIRPS data present maximum precipitation over the northwestern Amazon basin, central Brazil, and the SACZ (Figure 6a). Conversely, in the central Andes the accumulated precipitation is below 100 mm month-1. Figure 6a also shows regions of maximum precipitation over the eastern slopes of the Peruvian Altiplano known as hot-spots [70,71]. The RegGFDL simulation reproduces the precipitation pattern of CHIRPS data but overestimates it over the central Peruvian Amazon and northeastern Brazil (NEB) (Figure 6b). Additionally, Figure 6b displays that the RegGFDL simulation presents a slight underestimation of precipitation over the western and central parts of the equatorial Amazon basin. The RegMPI simulation reproduces the CHIRPS precipitation pattern but underestimates it over the western equatorial Amazon and SACZ (Figure 6c). Figure 6c also shows that the RegMPI overestimates the precipitation over NEB. Finally, the RegHadGEM2 simulation reproduces the precipitation pattern of CHIRPS data (Figure 6d), but the region of maximum precipitation over the continent occupies a smaller area compared to CHIRPS data. Figure 6d also indicates that the RegHadGEM2 simulation underestimates precipitation over the western equatorial Amazon. A common error in the three RegCM4 simulations is the overestimation of precipitation along the eastern slopes of the Peruvian and Bolivian Andes and the northern central Andes (Figure 6b–d).

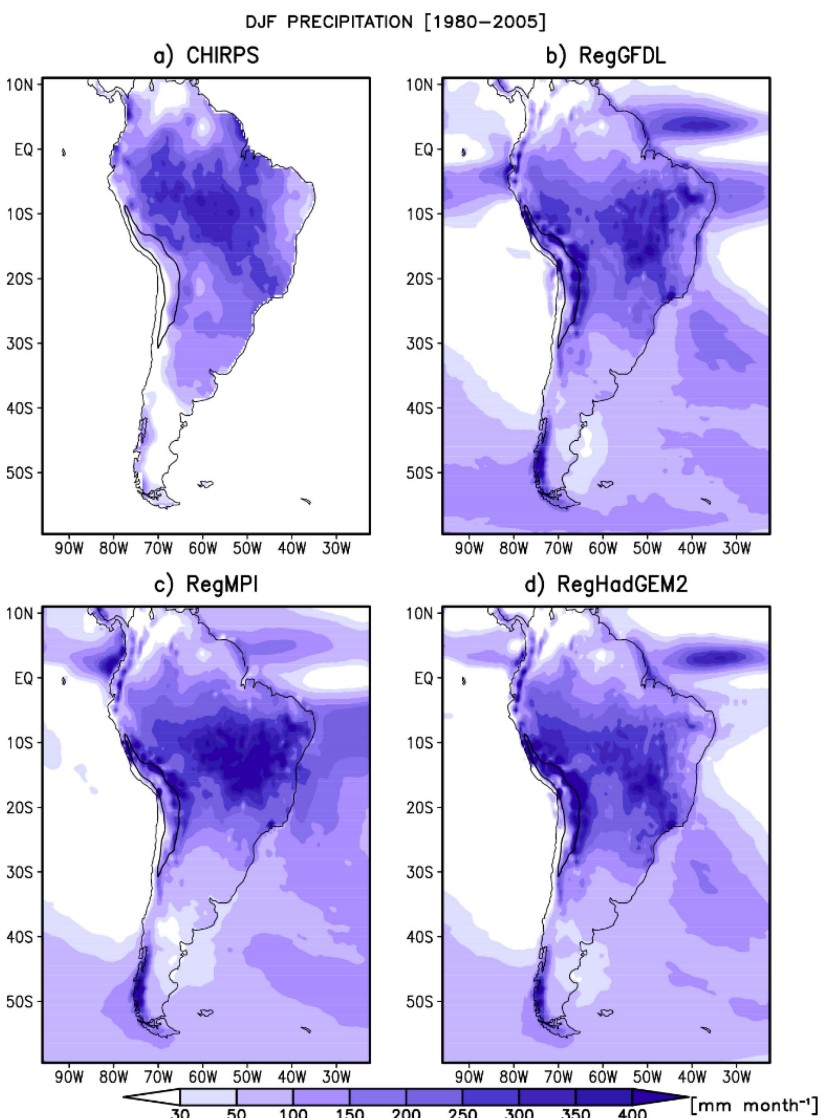

**Figure 6.** Climatology of DJF precipitation (mm month$^{-1}$) for the historical period (1980–2005): (**a**) CHIRPS, (**b**) RegGFDL, (**c**) RegMPI, and (**d**) RegHadGEM2 simulations.

The precipitation patterns are consistent with their respective 200 hPa circulation over South America (Figure 7). The three RegCM simulations reproduce the Bolivian high at 200 hPa, but it is located southwest of its observed position as a response to the overestimation of the precipitation over the central Andes and its eastern slope (Figure 7a–c). The overestimation of the precipitation over the NEB in the RegMPI simulation is caused by the northeastward shift of the Nordeste Low (Figure 7b). These comparisons show that the RegGFDL and RegHadGEM2 simulations reproduce better the Bolivian high-Nordeste low system over South America at 200 hPa than the RegMPI simulation.

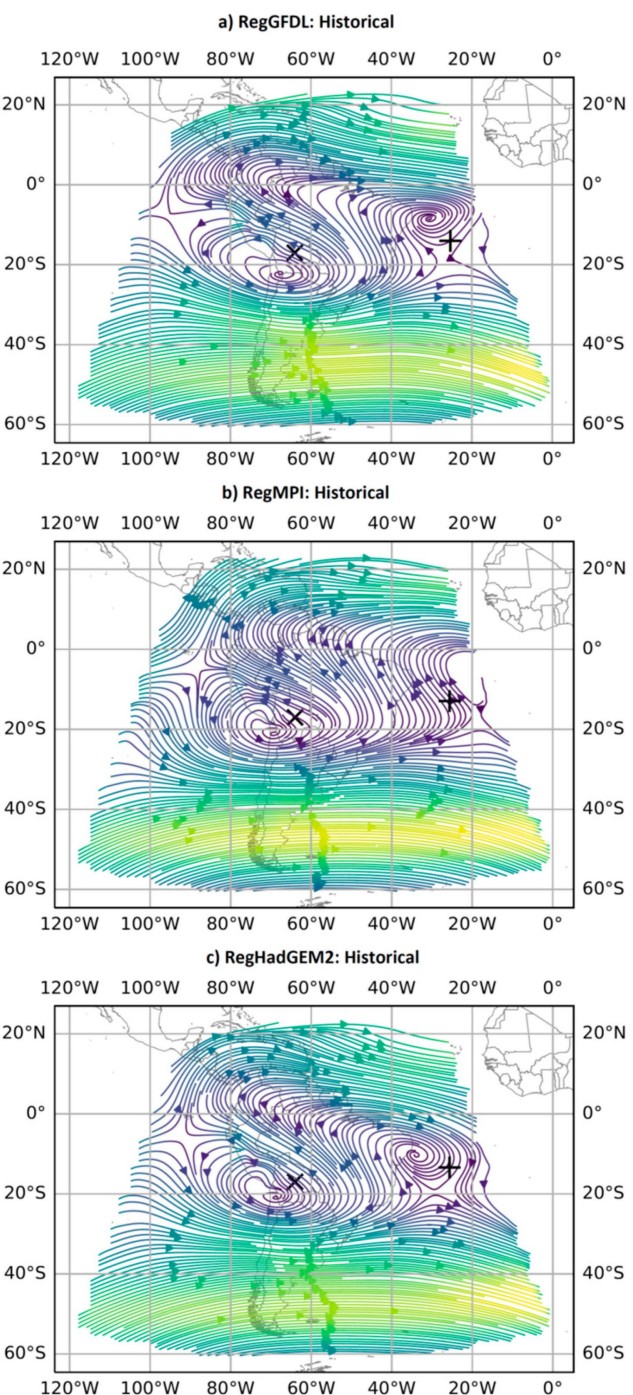

**Figure 7.** Climatology of DJF streamlines at 200 hPa for the historical period (1980–2005): (**a**) Reg-GFDL, (**b**) RegMPI, and (**c**) RegHadGEM simulations. The black X (cross) represents the position of the Bolivian high (Nodeste low) in the reanalysis.

### 3.3. Projected Changes in Precipitation and Circulation over the Equatorial Pacific Ocean and South America

For groups A, B, and C, Figure 8 presents the multimodel mean absolute differences (2070–2099 future minus 1980–2005 present climates) in precipitation, OLR, and 850 and 200 hPa winds over the equatorial Pacific Ocean and South America during the austral summer (DJF).

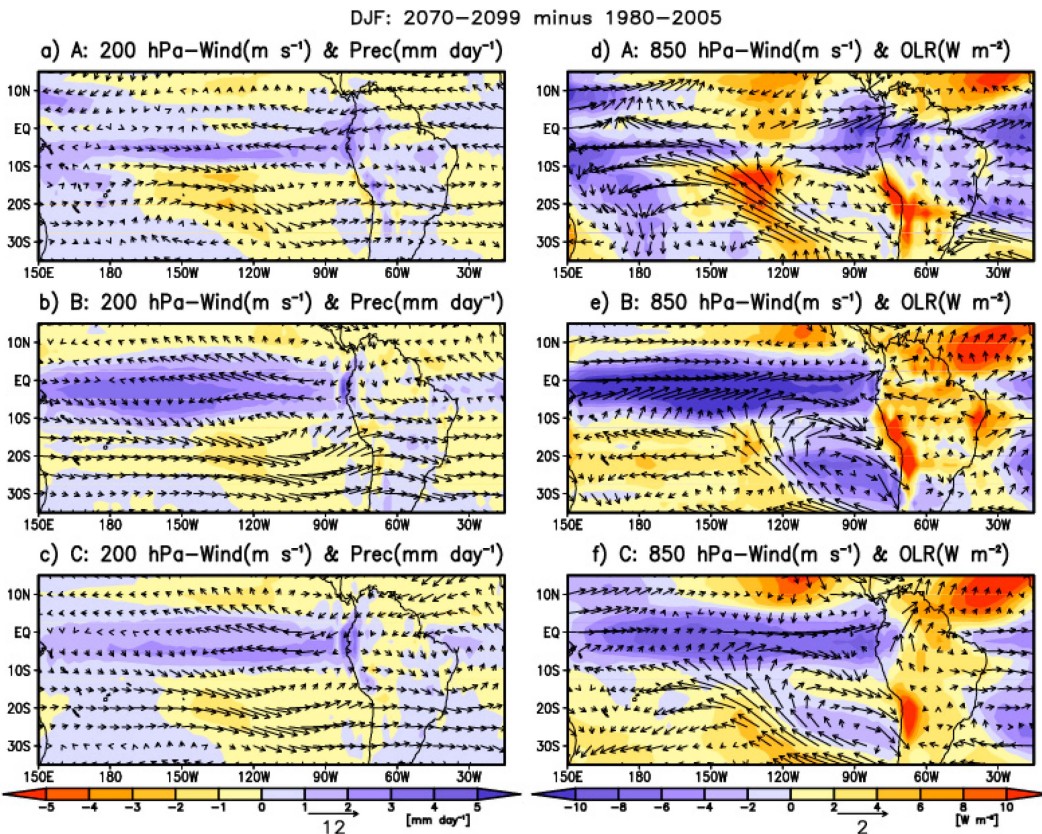

**Figure 8.** The multi-model ensemble of the absolute difference in the (**a**–**c**) 200 hPa wind (m s$^{-1}$) and precipitation (mm day$^{-1}$) and (**d**–**f**) 850 hPa wind (m s$^{-1}$) and OLR (W m$^{-2}$) between the 2070–2099 period and 1980–2005 period under the RCP 8.5 scenario for (**a**,**d**) group A (nonlinear ENSO characteristics and SACZ), (**b**,**e**) group B (nonlinear ENSO characteristics) and (**c**,**f**) group C (cannot simulate nonlinear ENSO characteristics).

Group A projects positive precipitation trends over the entire equatorial Pacific Ocean between 10° S and 0° N (Figure 8a). This projected increase in precipitation agrees with the projected increase in SST over the equatorial Pacific Ocean under the global warming scenario [72]. Conversely, absolute decreases in precipitation prevail in both hemispheres east of 170° W. The trends of the precipitation pattern agree with its corresponding large-scale upper-level circulation. There is an anticyclonic pair straddling the equator where the northern and southern nuclei are at approximately 90° W–5° N and 115° W–12.5° S, respectively. Groups B and C also project increases in precipitation over the entire equatorial Pacific Ocean but extend in a wider latitude band between 15° S and 5° N (Figure 8b,c). Figure 8b shows that group B projects positive precipitation trends over the central Pacific Ocean, which are stronger than those in groups A and C.

Groups B and C also feature an anticyclonic pair straddling the equator where the northern and southern nuclei are at approximately 100° W–7.5° N and 110° W–12.5° S, respectively, but these are less extensive than those in group A (Figure 8a–c). This means that more intense convective activity over the far-eastern Pacific is projected in group A than in groups B and C. Conversely, group A projects weaker warm conditions over the entire central Pacific Ocean than groups B and C.

In South America, groups A, B, and C project increases in precipitation over all of Ecuador (Figure 8a–c), consistent with the southwesterly low-level wind over the far-eastern Pacific associated with stronger convection due to the greater frequency of Eastern El Niño episodes [73]. All CMIP5 GCM groups project positive trends of precipitation over the western Amazon (Figure 8a–c). However, CMIP5 GCM groups project different patterns of OLR and low-level wind in the future (Figure 8d–f). For instance, groups A and C project a southwesterly low-level wind and a decrease in OLR over equatorial South

America (Figure 8d,f). In contrast, group B projects easterly low-level wind and increases in OLR over equatorial South America (Figure 8b). The representation of the SACZ pattern can explain the difference between groups A and B because the decrease in OLR in group A expands toward southeastern Brazil (Figure 8a).

Group A projects a positive precipitation trend over most of the central Andes, except for the southernmost sector of the central Andes, where decreases in precipitation prevail (Figure 8a). Figure 8a also shows the predominance of westerly differences in 200 hPa zonal wind over the entire central Andes but weakening toward the north. The upper-level westerly difference is associated with the projection of a weakened BH in the far future (Figure S1b). These trends show that the relationship between reduced precipitation and the westerly difference in 200 hPa zonal wind will only be maintained over the southern part of the central Andes, southward of 18° S. Figure 8d shows a positive trend of OLR and southeasterly differences in the low-level wind over the central Peruvian Amazon, located on the eastern slopes of the Peruvian Andes. These patterns suggest that the projected southeasterly difference in low-level winds over the central Peruvian Amazon favors stratiform precipitation episodes over the Peruvian Andes [70]. This hypothesis is consistent with the predominance of the projected reduction in OLR over most of the central Andes and the central Peruvian Amazon, suggesting a reduction of convective precipitation in this region in the far future (2070–2099).

Analogously, group C projects patterns of decreases in precipitation and westerly 200 hPa winds over the entire central Andes that are quite similar to those in group A but weaker than those in group A (Figure 8c and Figure S1e). These results suggest that CMIP5 GCMs in group C present incorrect compensatory feedback. These incorrect compensatory feedbacks would explain the high statistical scores (e.g., Pearson coefficient correlation, root-mean-square deviation, and coefficient of variation) between rain-gauge stations and precipitation simulated by five CMIP5 GCMs belonging to group C [46]. However, the description of the mechanism associated with this incorrect compensatory feedback is beyond the scope of the present study.

Conversely, group B projects a negative precipitation trend and westerly 200 hPa zonal wind over the entire central Andes, but these westerly differences are larger than those in groups A and C (Figure 8b). The upper-level westerly differences are associated with a weakened BH, and it even presents a northwestward displacement (Figure S1f). These projections are consistent with the patterns of the increase in OLR and the northeasterly low-level wind over the central Peruvian Amazon (Figure 8e). The low-level northeasterly projected differences are weaker in group B than in group A, which might be associated with the low performance of group B in the simulation of the SACZ.

We repeated the same analysis using three RegCM simulations to establish the change of the Bolivian high in the 2099–2070 period with respect to the 1980–2005 period under the RCP 8.5 scenario (Figure 9). The RegGFDL simulation depicts an anomalous anticyclonic circulation centered at 66° W, 18° S (Figure 9a), which indicates a strengthening and northward displacement of the Bolivian high (Figure S2a). Conversely, RegMPI and RegHadGEM project at 200 hPa westerly differences over the entire Andes of Peru and Bolivia (Figure 9b,c), which is an indication of future weakening of the Bolivian high (Figure S2b,c). These results highlight that the Bolivian high response under the RCP 8.5 scenario varies according to the group of CMIP5 GCMs, pointing out the existence of a great uncertainty of the response of the Bolivian high under the greenhouse gas future forcings.

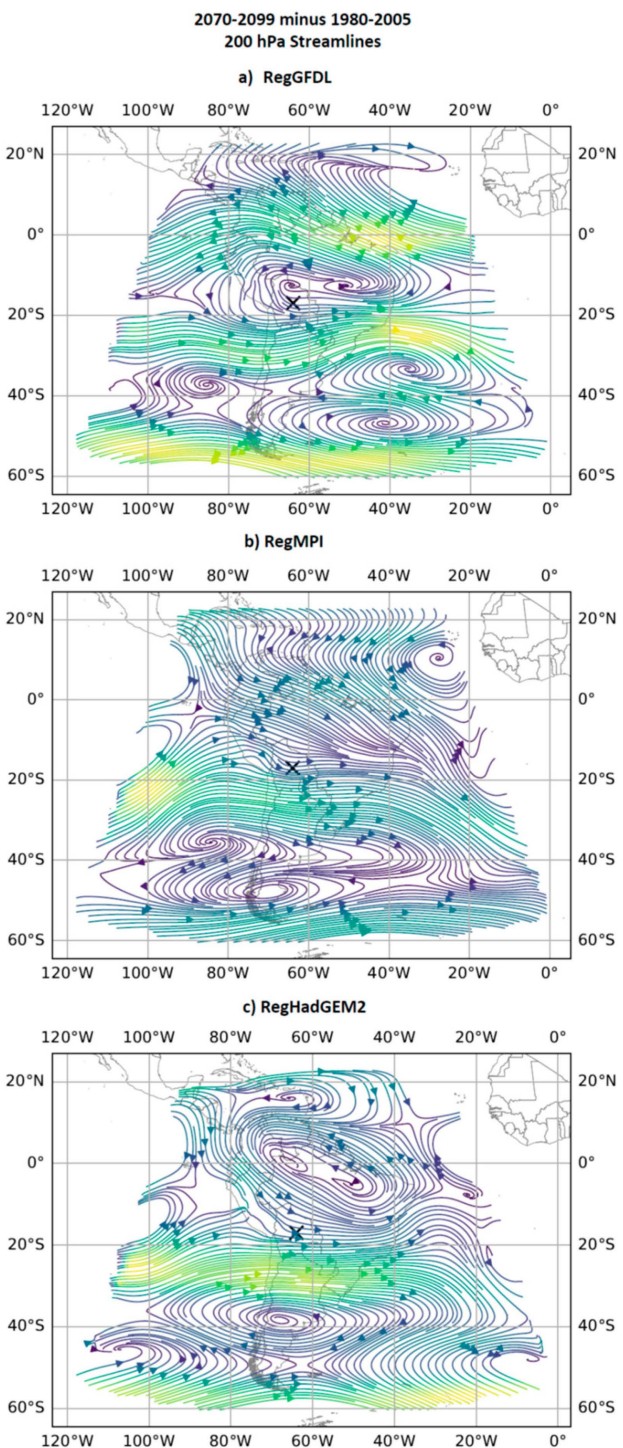

**Figure 9.** The DJF difference (2070–2099 minus 1980–2005) of the 200 hPa streamlines under the RCP 8.5 scenario for: (**a**) RegGFDL, (**b**) RegMPI, and (**c**) RegHadGEM. The RegHadGEM2 simulation covers the 2070–2098 period. The black cross represents the position of the Bolivian high in the historical climate (1980–2005).

### 3.4. Projected Trends of Rainfall in the Central Andes

Since the coarse resolution of all CMIP5 GCMs limits the quantification of the DJF precipitation trends over the entire central Andes, we also evaluated the future rainfall trends using regional climate projections (RegGFDL, RegMPI, and RegHadGEM) (Figure 10).

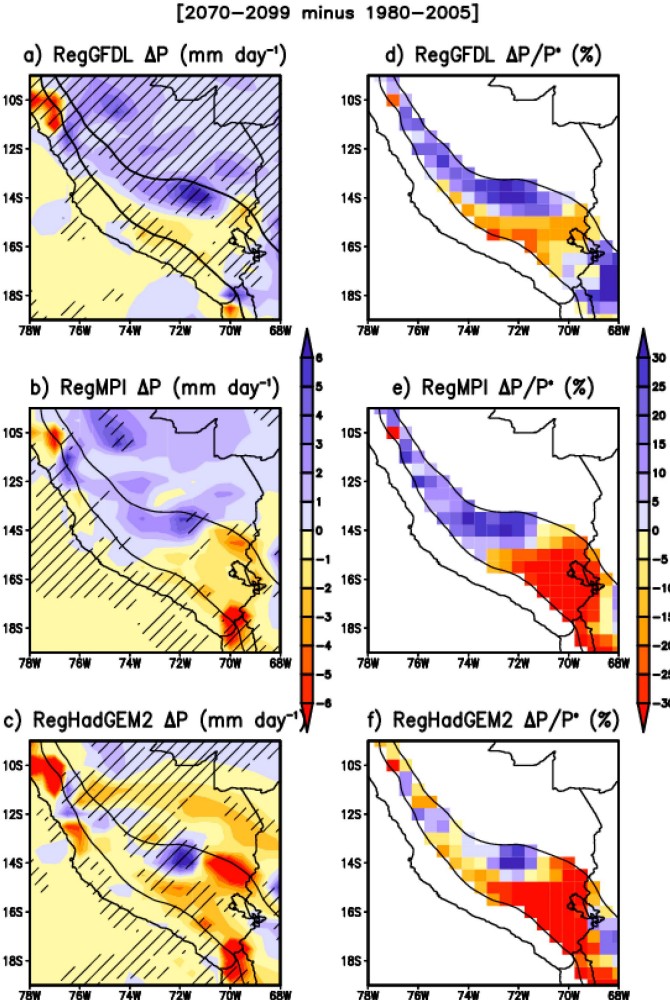

**Figure 10.** (**a–c**) Absolute difference in DJF precipitation (mm day$^{-1}$) for the central Andes between 2070–2099 and 1980–2005 under the RCP 8.5 scenario for (**a**) RegGFDL, (**b**) RegMPI, and (**c**) RegHadGEM simulations. (**d–f**) As in (**a–c**) but for the relative difference in DJF precipitation (in %). The thin black line represents the 3000 m topographic contour. Hatching indicates a statistically significant difference in the mean at the 90% confidence level.

For the absolute trend (2070–2099 minus 1980–2005), the RegGFDL and RegMPI project increases in precipitation of more than +3 mm day$^{-1}$ over the northern-central Andes and the central Amazon basin. However, the projected rainfall changes are only statistically significant at 90% in the RegGFDL (Figure 10a,b). This further indicates that climatological features of DJF precipitation over the central Andes and the central Peruvian Amazon will change in the far future (2070–2099).

Conversely, all RegCM4 simulations project absolute decreases in precipitation over the central and southern-central Andes, approximately −1 mm day$^{-1}$ (Figure 10a–c). These projections are consistent with previous studies based on the statistical downscaling of CMIP3 models [11] and agree with RegCM3-CMIP3 projections [47] and the multimodel ensemble of RCM simulations from CORDEX [48]. Recently, low- and high-resolution experiments of the RegCM4 model, which was forced with the MPI-ESM-MR model, project a reduction of summer precipitation (below −5 mm day$^{-1}$) over the northern Chilean Andes (e.g., the southernmost part of the central Andes) for the 2021–2050 period under the RCP 2.6 and 8.5 scenarios [74]. Figure 10c also shows that the RegHadGEM simulation projects a statistically significant reduction in precipitation over the central Peruvian Amazon. The sign of the projected absolute precipitation difference over the entire central Andes is consistent with the long-term trends of DJF precipitation in the

CMIP5 GCMs and the RegCM4 simulations (Figure 11). A consistent positive trend in precipitation of approximately +0.003 yr$^{-1}$ projected in the RegGFDL is noted (Figure 11a). In comparison, the RegMPI and RegHadGEM2 project negative trends ($-0.003$ yr$^{-1}$ and $-0.02$ yr$^{-1}$, respectively; Figure 11b,c). Moreover, the intensity and sign of the trends in the RegGFDL and RegMPI simulations are consistent with the trends of their respective CMIP5 GCMs forcing while the opposite occurs with the RegHadGEM2 simulation and its respective GCMs (Figure 11).

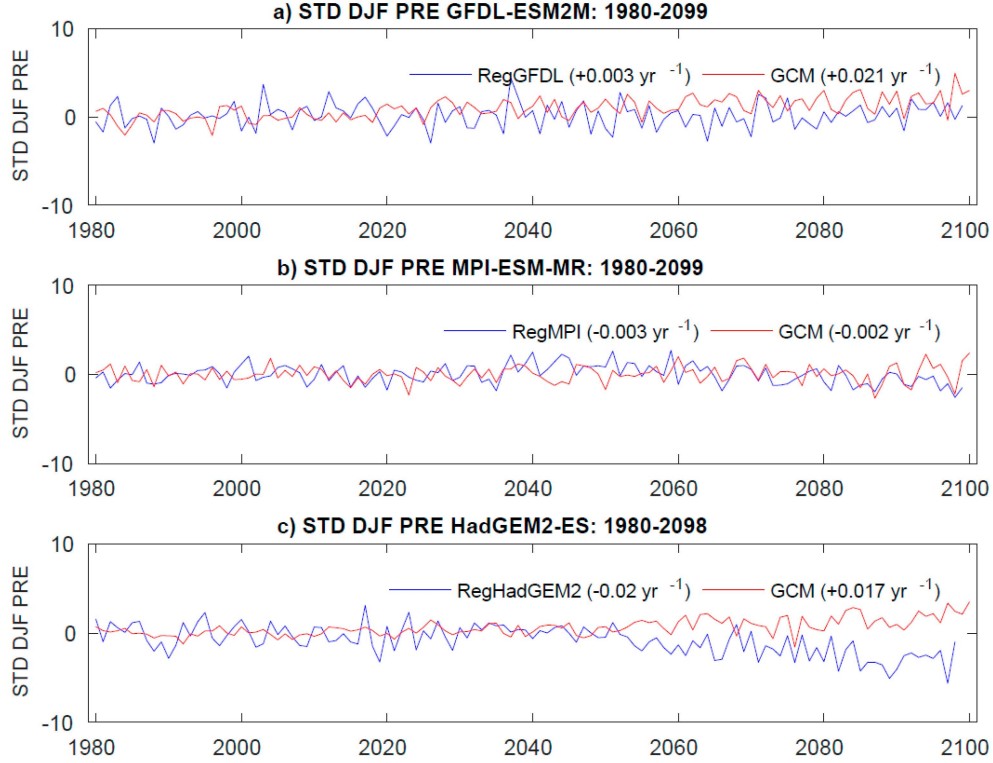

**Figure 11.** The trend of standardized anomalies of the average DJF precipitation (mm day$^{-1}$) in the entire central Andes for the GCM (red line) and RegCM4 simulations (blue line) of (**a**) GFDL-ESM2M, (**b**) MPI-ESM-MR, and (**c**) HadGEM2-ES. The black lines represent the linear regression fits.

Concerning the relative trends (in %), the RegGFDL projects a relative increase, between 10% and 15%, over the northern and central Andes, while in the rest of the Andes (south of 13°S) it projects a relative decrease, between $-15$% and $-5$% (Figure 10d). Figure 7e shows that the RegMPI projects a relative increase, between 5% and 15%, over the central and northern-central Andes. In the southern-central Andes, the projections indicate a negative trend of approximately $-20$%. Conversely, the RegHadGEM projects a relative decrease in precipitation over most of the central Andes, between $-25$% and $-5$% (Figure 10f). Therefore, RegCM4 simulations forced by CMIP5-GCM group C (RegMPI and RegHadGEM2) do not project the same precipitation trend over the northern-central Andes and the central Peruvian Amazon, indicating that GCMs have a key role in the precipitation projections in these regions.

Although the RegGFDL projection agrees with the projected changes in DJF precipitation over the entire central Andes in the CMIP5 GCMs from group A, the uncertainty is still great since it is based on single RegGFDL experiment due to the unique GFDL-ESM2M simulation under the RCP 8.5 scenario.

## 4. Discussion and Conclusions

Only eight (CMCC-CMS, CMCC-CESM, GISS-E2-R, GFDL-ESM2M, BNU-ESM, CNRM-CM5, GFDL-CM3and CCSM4) out of 27 CMIP5 GCMs simulate simultaneously the nonlin-

ear ENSO parameter "α" and the nonlinear behavior of the central and eastern El Niño. These results point that these eight CMIP5 GCMs reproduce the mean-state of the Walker cell over the Pacific and the local Walker cell over South America as well as the change of the intensity and the structure of the Pacific Walker cell during the extreme episodes of the central and eastern El Niño [29]. The ability of models to simulate the nonlinear ENSO characteristics reduces the uncertainty in the reproduction of the atmospheric teleconnection between different types of El Niño and austral summer precipitation over the central Andes [15,16].

Most CMIP5 GCMs cannot simulate the main features of the regional atmospheric circulation and precipitation over South America during the austral summer. Only three out of 21 CMIP5 GCMs (GFDL-ESM2M, CCSM4, and BNU-ESM) simulate the nonlinear ENSO characteristics and the SACZ. The poor performance in simulating the SACZ is related to the poor performance to simulate the atmospheric teleconnection between the SACZ and precipitation in the equatorial Amazon basin. This further explains the simulation of a smaller extension of the BH (CMCC-CESM and GISS-E2-R) and NL southward of its observed position (CNRM-CM5). Therefore, the simulation of the atmospheric teleconnection between the SACZ and equatorial Amazon basin favors the correct simulation of the SACZ and the local Walker cell over the South American continent.

The simultaneous simulation of the nonlinear ENSO characteristics and the SACZ evidences the existence of the SACZ-ENSO coupling. This coupling is a new framework to describe the climatological characteristics of the regional atmospheric circulation and precipitation over South America and even how it will change in the future. However, this new feature is simulated only by three out of 21 CMIP5 GCMs, thereby showing the limitations of the current CMIP5 GCMs.

There is an agreement in all CMIP5 GCM groups as to the projection of an increase in DJF precipitation over the eastern Pacific Ocean. This occurs because of the projection of an increase in extreme episodes of the Eastern El Niño for the end of the 21st century in most CMIP5 GCMs [73]. Conversely, there is great uncertainty in the DJF precipitation projection over the central Pacific Ocean.

All CMIP5 GCM groups also project an absolute trend in the westerly zonal wind over the central Andes at 200 hPa and in the southeasterly low-level wind over the central Peruvian Amazon, but the differences are larger in group A. Conversely, CMIP5 GCMs project different patterns of DJF precipitation trends over the central Andes. For instance, group A projects increase in DJF precipitation over the central Andes, while group B projects a decrease. The difference between groups A and B is related to the simulation of the SACZ associated with the simulation of the low-level wind over the central Peruvian Amazon. Moreover, all CMIP5 GCM groups project increases in DJF OLR over the entire central Andes, indicating a reduction in deep convection activity in this region and the central western Amazon.

RegCM4 simulates the main features of the precipitation patterns, South América low-level jet and the upper-level circulation streamlines over South America during the austral summer. However, all RegCM4 experiments simulate the Bolivian high southwestward displaced from its observed position (e.g., NCEP/NCAR reanalysis), which is related to the overestimation of the precipitation (and consequent latent heating release) over the central Andes and along of its eastern slopes. The overestimation of the precipitation is, in part, associated with the underestimation of the topography elevation, which even occurs using 50 km of grid spacing in RegCM4.

Due to the coarse resolution of CMIP5 GCMs, we use RegCM4 simulation to evaluate the projected changes in DJF precipitation over the central Andes. The RegGFDL simulation projects a relative increase in precipitation over most of the central Andes, which varies between 5% and 15%. Conversely, a relative decrease (of approximately −10%) in precipitation is projected in the central and southern parts of the central Andes. This projected reduction of precipitation agrees with the projections of statistical downscaling ESD and fine-resolution RegCM4 simulations [11,74].

There is great uncertainty in estimating the projected changes in precipitation and low- and upper-level wind over South America during the austral summer, which is related to the limited number of CMIP5 GCMs that simulate the nonlinear ENSO characteristics and the SACZ during this period. For example, the CCSM4 model has six simulations under the RCP 8.5 scenario, while the GFDL-ESM2M and BNU-ESM models have only one simulation for the same scenario. This constraint also limits the RegCM4 simulations of present and future South American climates, particularly for the central Andes, which presents complex topography.

The uncertainty in the projection of the absolute difference in DJF precipitation over the central Andes is also related to an incorrect feedback simulation in the Andes-Amazon region observed in most CMIP5 GCMs, e.g., group C, which contains 19 out of 25 CMIP5 GCMs. Even having incorrect feedback, the group C realistically simulates the climatological features of the austral summer precipitation over the central Andes [46].

Finally, these results will help in the selection of CMIP5 GCMs as input to study the impacts of climate change on precipitation and temperature in the tropical Andes and Amazon basin by running high-resolution mesoscale models such as the Weather Research and Forecasting (WRF) model.

**Supplementary Materials:** The following are available online at https://www.mdpi.com/article/10.3390/cli9050077/s1. Figure S1: Climatology of DJF precipitation (mm day-1) and 200 hPa wind (stream) over the Pacific basin and South America for (a–c) present (1980–2005) and (d-f) future (2070–2099) climates under the RCP 8.5 scenario for groups (a,d) A (nonlinear ENSO characteristics and SACZ), (b,e) B (nonlinear ENSO characteristics), and (c,f) C (cannot simulate nonlinear ENSO characteristics). Figure S2. Climatology of DJF 200 hPa circulation over South America for the future climate (2070–2099): (a) RegGFDL, (b) RegMPI, and (c) RegHadGEM simulations. The RegHadGEM2 simulation covers the 2070–2098 period. The black cross represents the position of the Bolivian high in the historical climate (1980–2005).

**Author Contributions:** Conceptualization, J.C.S.; simulations, R.P.d.R. provides the RegCM4 simulations, methodology, J.C.S., and R.P.d.R.; analysis, J.C.S., and R.P.d.R.; writing—original draft preparation; J.C.S. and R.P.d.R. writing the response to reviewers. Both authors have read and agreed to the published version of the manuscript.

**Funding:** This research was funded by Peruvian PPR068 program "Reducción de vulnerabilidad y atención de emergencias por desastres".

**Data Availability Statement:** The reanalysis data are provided at NCEP/NCAR (https://psl.noaa.gov/data/gridded/data.ncep.reanalysis.html; access date: 1 August 2020). OLR data are available at NCEP/NOAA (https://psl.noaa.gov/data/gridded/data.interp_OLR.html; access date: 1 January 2020). GPCP data was provided by Global Precipitation Climatology Project (http://www.esrl.noaa.gov/psd/data/gridded/data.gpcp.html; access date: 2 January 2020). The CMIP5 simulations may be downloaded from CEDA (https://data.ceda.ac.uk/badc/cmip5/data/cmip5/output1/ with access permission; access date: 1 July 2019). The three RegCM4 simulations are not available publicly. Written correspondence can be sent to Rosmeri Porfirio da Rocha (rosmerir.rocha@iag.usp.br) to obtain these RegCM4 simulations.

**Acknowledgments:** Juan Sulca was funded by Peruvian PPR068 program "Reducción de vulnerabilidad y atención de emergencias por desastres". The work was performed using computational resources, HPC-Linux-Cluster, from Laboratorio de Dinámica de Fluidos Geofísicos Computacionales at Instituto Geofísico del Perú (grants 101-2014-FONDECYT). R. P. da Rocha thanks to CNPq-Brazil (#304949/2018-3 and 430314/2018-3). Moreover, the authors thank Enciso for his help with the production of some figures. We are very grateful to three anonymous reviewers who provided us with valuable comments, which helped us to advance our results significantly and to improve the paper.

**Conflicts of Interest:** The authors declare no conflict of interest.

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
