# Peer review of "Influence of the Coupling South Atlantic Convergence Zone-El Niño-Southern Oscillation (SACZ-ENSO) on the Projected Precipitation Changes over the Central Andes"

_climate, doi:10.3390/cli9050077_

Round 1
Reviewer 1 Report
Review for “Influence of the South Atlantic Convergence Zone-ENSO pattern variability on the projected precipitation changes over the central Andes”
This research uses the CMIP5 GCMs and RegGFDL model to understand ENSO variability from the GCM and how it influences the precipitation pattern over the central Andes. Although there is a great amount of uncertainties shown by the results, the work itself is solid and the paper is well written. Therefore, I recommend a minor revision for this research and the authors need to address the issues shown as below.
Line 55. MSL is used without definition and “m.a.s.l” is used in Figure 2. Please keep consistent.
Line 67 [15] found, please include the author of the literature.
Line 75, same for [9]
Line 253 to 257: Why three GCMs do not reproduce the nonlinear behavior of ENS are included in the remaining analysis?
Author Response
Review for “Influence of the South Atlantic Convergence Zone-ENSO pattern variability on the projected precipitation changes over the central Andes”
This research uses the CMIP5 GCMs and RegGFDL model to understand ENSO variability from the GCM and how it influences the precipitation pattern over the central Andes. Although there is a great amount of uncertainties shown by the results, the work itself is solid and the
paper is well written. Therefore, I recommend a minor revision for this research and the authors need to address the issues shown as below.
Line 55. MSL is used without definition and “m.a.s.l” is used in Figure
2. Please keep consistent.
We deleted the term “m.a.s.l”.
Line 67 [15] found, please include the author of the literature.
We rephrased the sentence (lines 68-75).
The positive central El Niños inhibit precipitation over the central Andes and the equatorial Amazon east of 70°W. Concerning positive Eastern El Niño episodes, the warm conditions over the far-eastern Pacific induce precipitation over the coasts of southern Ecuador and northern Peru while inhibiting precipitation over the Peruvian Andes south of 8°S. However, the reduction of precipitation over the Peruvian Andes is only statistically significant over the Peruvian Altiplano. Conversely, both Central and Eastern La Niñas induce a precipitation increase over the central Andes [15].
Line 75, same for [9]
We rephrased the sentence (lines 75-76).
Moreover, ENSO is an essential modulator of the upper-level zonal wind over Peru during austral summer [9].
Line 253 to 257: Why three GCMs do not reproduce the nonlinear behavior
of ENSO are included in the remaining analysis?
The addition of these three last CMIP5 GCMs is because Sulca (2021) did not assess the performance of the simulation of the nonlinear ENSO characteristics in these three models. The addition of the last three CMIP5 GCMs improves the robustness of the composite of group C.
Reviewer 2 Report
The authors study current and future climate as simulated by CMIP5 GCMs under RCP8.5. They grouped the GCMs according to their ability to simulate the SACZ and ENSO effects. They also use the RegCM4 to downscale three GCMs. The authors conducted a very elaborate study, which is interesting, however the manuscripts needs further work. I recommend major revisions because: In section 2.1 All the listed datasets ("observations") are course resolution, and will not be good enough to compare against the RegCM4 simulations. Section 3.3: The performance of the RegCM4 nested within the three GCMs when simulating present day climate is not presented, the downscaling component jumps straight to projections. The authors do not compare the RegCM4 projections to the individual GCMs used to the force the regional model. The authors do not cite other studies that have done similar work and found limitations in the performance of CMIP5 models, and they also don't compare their projections with others in the literature. Their section 4 hardly cites other studies. Minor revisions The authors introduce the use of Figures in the Introduction stage. The disadvantage with this is that the Figures are presented with the reader unaware of what the source of the data is, that is used to produce the Figures. I suggest that these be introduced only once the data used in the study has been discussed. Where authors start a sentence with a reference or use in the middle of the sentence, they don’t mention the surname of the lead author of the paper. They only list a number. Line 94 is a one sentence paragraph. Please merge with previous paragraph. Line 99: Replace “but only that over the centra …was statistically significant” with but that the increase was statistically significant only over the central Peruvian Amazon. Line 104 to 105 – a number of models listed without being written in full Line 129: there are two dynamical downscaling techniques I am aware of, stretched grid to give higher resolution over the area of interest, or the traditional nesting method. I suggest replacing “several dynamical downscaling techniques were” with “a dynamical downscaling technique was” Line 30: replace as with at Line 131: before mentioning projections from regional models, say something about their performance Line 143: Write CORDEX in full when mentioning it for the first time Line 140: Define RCP and mention briefly what RCP8.5 represents Line 217 is a repeat of line 211 to 212. Merge the naming convention to line 212 and delete line 217. Line 202- add a plot showing the domain and topography of the domain…Fig 2 can be modified for this purpose. Line 262 and 262 – There is no Figure 2a and 2b Line 266- analysis or reanalysis? Line 320 – 321: cite literature that have found that an underestimation of the height of mountains and parametrization deficiency results in rainfall overestimation in GCMs. Line 402: replace reducing with “a reduction of”Author Response
The authors study current and future climate as simulated by CMIP5 GCMs under RCP8.5. They grouped the GCMs according to their ability to simulate the SACZ and ENSO effects. They also use the RegCM4 to downscale three GCMs. The authors conducted a very elaborate study, which is interesting, however the manuscripts needs further work. I recommend major revisions because:
Thank you for your comments.
We rewrote the title and reorganized the text following your suggestions.
Major revisions
In section 2.1 All the listed datasets ("observations") are course resolution and will not be good enough to compare against the RegCM4 simulations.
We agree.
We added subsection 3.2, “Performance of the RegCM4 model”.
We compared the precipitation field simulated by the RegCM4 simulation with the rescaled precipitation data from CHIRPS (Fig. 6; lines 441-458). We also added new figures of the 200 hPa circulation over South America to describe the response of the Bolivian High-Nordeste low system under the RCP 8.5 scenario (Fig. 7, lines 466-474).
Section 3.3: The performance of the RegCM4 nested within the three GCMs when simulating present day climate is not presented, the downscaling component jumps straight to projections.
Yes, you are right.
We added subsection 3.2, “Performance of the RegCM4 model” (lines 426-474).
The authors do not compare the RegCM4 projections to the individual GCMs used to the force the regional model.
Thank you for your observation.
We added information on the future precipitation projection of the GFDL-ESM2M, MPI-ESM-MR, and HadGEM2-ES to compare with their respective RegCM4 simulation (Fig. 11, lines 615-618).
The authors do not cite other studies that have done similar work and found limitations in the performance of CMIP5 models, and they also don't compare their projections with others in the literature.
Thank you for your observations.
We added new references to amend this problem.
However, few specific studies exist about the future precipitation projection over the central Andes using the RegCM4 model.
Minvielle, M., Garreaud, R.D., 2011. Projecting rainfall changes over the South American Altiplano. J. Climate. 24(17), 791 4577-4583. http://doi.org/10.1175/JCLI-D-11-0005.11
Centro de Ciencia del Clima y la Resilencia (CR)2, 2018. Simulaciones climáticas regionales y marco de evaluación de la 951 vulnerabilidad. Ministerio del Medio Ambiente de Chile. www.cr2.cl. (archived on 01 April 2021).
Section 4 hardly cites other studies.
Thank you for your observations.
We added new references to amend this problem.
Sulca, J., 2021. Evidence of nonlinear Walker circulation feedbacks on extreme El Niño Pacific diversity: observations and 830 CMIP5 models. Int. J. Climatol. 41, 2934-2961. https://doi.org/10.1002/joc.6998
Abadi, A.M., Oglesby, R., Rowe, C., Mawalagedara, R., 2018. Evaluation of GCMs historical simulation of monthly and 845 seasonal climatology over Bolivia. Clim. Dyn. 51, 733-754. https://doi.org/10.1007/s00382-017-3952-y
Zubieta, R., Molina-Carpio, J., Laqui, W., Sulca, J., Ilbay, M., 2021. Comparative Analysis of Climate Change 874 Impacts on Meteorological, Hydrological, and Agricultural Droughts in the Lake Titicaca Basin. Water, 13, 875 175. https://doi.org/ 10.3390/w13020175
Centro de Ciencia del Clima y la Resilencia (CR)2, 2018. Simulaciones climáticas regionales y marco de evaluación de la 951 vulnerabilidad. Ministerio del Medio Ambiente de Chile. www.cr2.cl. (archived on 01 April 2021).
Minor revisions
The authors introduce the use of Figures in the Introduction stage. The disadvantage with this is that the Figures are presented with the reader unaware of what the source of the data is, that is used to produce the Figures. I suggest that these be introduced only once the data used in the study has been discussed.
Thank you for your suggestion.
We changed the positions of Figures 1 and 2.
Where authors start a sentence with a reference or use in the middle of the sentence, they don’t mention the surname of the lead author of the paper. They only list a number.
Thank you for your observation.
We rephrased the paragraphs that present this mistake.
Line 94 is a one sentence paragraph. Please merge with previous paragraph.
Yes, you are right.
We added more information to the original sentence to complete a paragraph (lines 104-108).
Line 99: Replace “but only that over the centra …was statistically significant” with but that the increase was statistically significant only over the central Peruvian Amazon.
Thank you for the observation and suggestion.
It was done (line 125).
Line 104 to 105 – a number of models listed without being written in full
Thank you for the observation.
The list of models was fixed (Table 1).
Line 129: there are two dynamical downscaling techniques I am aware of, stretched grid to give higher resolution over the area of interest, or the traditional nesting method. I suggest replacing “several dynamical downscaling techniques were” with “a dynamical downscaling technique was”
We agree.
The sentence was fixed (line 167).
Line 30: replace as with at Line 131: before mentioning projections from regional models, say something about their performance
Thank you for your observation.
We added subsection 3.2, “Performance of the RegCM4 model” (lines 426-474).
Line 143: Write CORDEX in full when mentioning it for the first time
It was done (line 204).
Line 140: Define RCP and mention briefly what RCP8.5 represents
Thank you for the suggestions.
We added a paragraph in the Introduction (lines 154-162).
Line 217 is a repeat of line 211 to 212. Merge the naming convention to line 212 and delete line 217.
Thank you for the suggestions.
It was done (lines 279-282).
Line 202- add a plot showing the domain and topography of the domain…Fig 2 can be modified for this purpose.
Thank you for the suggestions.
It was done (new Fig. 4).
Line 262 and 262 – There is no Figure 2a and 2b
Thank you for the observation.
They must be Figures 1a and 1b (lines 334 and 337).
Line 266- analysis or reanalysis?
Thank you for the observation.
It must be reanalysis (line 340).
Line 320 – 321: cite literature that have found that an underestimation of the height of mountains and parametrization deficiency results in rainfall overestimation in GCMs.
Thank you for the observation.
We added Saavedra et al. (2020) to amend it (line 396).
Line 402: replace “reducing” with “a reduction of”
Thank you for the suggestion.
It was done (line 536).

Reviewer 3 Report
Review of Manuscript ID: climate-1166676, titled “Influence of the South Atlantic Convergence Zone-ENSO pattern variability on the projected precipitation changes over the central Andes”
The manuscript seeks to identify the strengths and shortcomings of 25 CMIP5 global climate models in simulating the nonlinear ENSO characteristics and the South Atlantic Convergence Zone (SACZ) in relation to the regional hydroclimate of central Andes. The authors use an EOF analysis of monthly gridded OLR data to obtain the SACZ pattern. Thereafter, they demonstrate the simulation and the projection of precipitation, OLR, 200-hPa and 850-hPa, winds by clustering these CMIP5 GCMs for the historical and end-of-century period respectively. The examination reveals that only three GCMs simulate the nonlinear characteristic of ENSO and the SACZ, and there is uncertainty in the projected changes of regional precipitation over South America.
Overall, the authors provide an interesting framework to diagnose this research problem. There are novel elements of interest to the readership of MDPI Climate. The high quality of the figures is noted. I thereby recommend to publish this manuscript after addressing a few comments of minor nature.
- The method of determination of the nonlinear characteristic of ENSO is unclear to the reader. Please expand the related discussion in section 2.4 to better describe how the metric, alpha, is determined and how the value should be interpreted. This is especially important because this methodology forms a pivotal part of the paper.
- Figs. 3, 4: The meaning of the numbers within parenthesis in the figure legend should be mentioned in the figure caption, i.e., they are pattern correlations between the OBS and each individual GCM.
- Fig. 5: It would probably help the reader follow the figure discussion better if the corresponding observed panel (from Fig. 1) is re-plotted here.
- Lines 160-165: Please check the numbering convention stated here, e.g., Section 2 is the data and methods in this manuscript, and not section 3, and, so on, and so forth.
Author Response
Review of Manuscript ID: climate-1166676, titled “Influence of the South Atlantic Convergence Zone-ENSO pattern variability on the projected precipitation changes over the central Andes”
The manuscript seeks to identify the strengths and shortcomings of 25 CMIP5 global climate models in simulating the nonlinear ENSO characteristics and the South Atlantic Convergence Zone (SACZ) in relation to the regional hydroclimate of central Andes. The authors use an EOF analysis of monthly gridded OLR data to obtain the SACZ pattern. Thereafter, they demonstrate the simulation and the projection of precipitation, OLR, 200-hPa and 850-hPa, winds by clustering these CMIP5 GCMs for the historical and end-of-century period respectively. The examination reveals that only three GCMs simulate the nonlinear characteristic of ENSO and the SACZ, and there is uncertainty in the projected changes of regional precipitation over South America.
Overall, the authors provide an interesting framework to diagnose this research problem. There are novel elements of interest to the readership of MDPI Climate. The high quality of the figures is noted. I thereby recommend publishing this manuscript after addressing a few comments of minor nature.
Thank you for your comments.
We rewrote the title and reorganized the text following your suggestions.
- The method of determination of the nonlinear characteristic of ENSO is unclear to the reader. Please expand the related discussion in section 2.4 to better describe how the metric, alpha, is determined and how the value should be interpreted. This is especially important because this methodology forms a pivotal part of the paper.
Thank you for your observation and suggestion.
We added some paragraphs about the method of determination of the nonlinear ENSO parameter “α” in the Introduction (lines 111-115). The physical interpretation of the parameter “α” is that the ENSO nonlinearity is associated with the intensity of the mean-state of the Pacific Walker cell (lines 117-119).
- Figs. 3, 4: The meaning of the numbers within parenthesis in the figure legend should be mentioned in the figure caption, i.e., they are pattern correlations between the OBS and each individual GCM.
Thank you for the observation.
The meaning of the numbers within parentheses was added in the captions of Figures 3 and 4.
- Fig. 5: It would probably help the reader follow the figure discussion better if the corresponding observed panel (from Fig. 1) is re-plotted here.
We agree.
We copied Figure 1 into Figure 5a.
- Lines 160-165: Please check the numbering convention stated here, e.g., Section 2 is the data and methods in this manuscript, and not section 3, and, so on, and so forth.
Yes, you are right.
The sequence of the sections was fixed (lines 221-225).

Reviewer 4 Report
Please see the attached pdf file.

Author Response
Reviewer #4
This study examines the projected precipitation changes over the central Andes in both the CMIP5 models and RegCM4. The topic is interesting. However, some improvements are needed in presenting the results. In particular, there are a lot of errors in describing figures and in figure captions. Therefore, I recommend rejection of this paper for publication.
Thank you for your comments. Following your and other reviewer suggestions we modified the manuscript to improve the presentation of our results. We rewrote the title and reorganized the text following your suggestions.
Major comments:
- It seems that the nonlinear ENSO characteristics are an important factor to classify the models into different groups. However, it is not clear what the nonlinear ENSO characteristics are. No results are provided in terms of models performance in simulating these nonlinear ENSO characteristics.
We added a new paragraph discussing the role of the nonlinear ENSO characteristics in the performance of the CMIP5 GCMs and summer precipitation over the central Andes (lines 78-85, 104-108, 111-119).
We also added information about the key role of the nonlinear ENSO characteristics in the section Results (lines 323-331).
- Title: SACZ-ENSO pattern variability is misleading. It sounds like a coupled SACZ-ENSO variability.
We changed the original title by “Influence of the coupling South Atlantic Convergence Zone-El Niño-Southern Oscillation on the projected precipitation changes over the central Andes”
- CMIP5: The CMIP5 model outputs are used in this study. However, CMIP5 is relatively old, and CMIP6 data are already available. It would be useful and informative to look at the CMIP6 results and compare them with the CMIP5.
Your observation and suggestion are remarkably interesting. However, the present study was planned to use regional and global models to discuss the impact of the coupling SACS-ENSO in the central Andes. At moment only are available regional simulations nested in CMIP5 GCMs. In future work, it will be interesting to analyze the CMIP6 GCMs to identify improvements or not compared to CMIP5.
- Lines 254-255: What is the relationship between the nonlinear behavior of ENSO and the nonlinear ENSO metric?
The relationship between the nonlinear behavior of ENSO and the nonlinear ENSO metric is detailed in the Introduction (lines 78-85, 104-108). The ability of the simultaneous simulations of the nonlinear ENSO characteristics in the CMIP5 GCMs guarantees the reproduction of the atmospheric teleconnection of ENSO associated with the austral summer precipitation over the central Andes (lines 327-331). We also refer to Sulca (2021) that gives more details of the nonlinear ENSO characteristics.
- Lines 305-313: The authors claim that the simulation of the SACZ is a great deficit in some CMIP5 GCMs based on the leading EOF of OLR. However, the EOF analysis is applied to OLR anomalies. Why not just look at the long-term mean precipitation in each model, which should be a good indicator for the mean SACZ?
The long-term mean precipitation provides some evidence of the performance of the CMIP5 GCMs to simulate the SACZ. However, this factor alone does not guarantee the coupling SACZ-ENSO. Thus, we analyzed the EOF pattern of DJF OLR anomalies that allows us to check the correct simulation of the atmospheric teleconnection of the SACZ with austral summer precipitation over the equatorial Pacific Ocean and South America. This concept was initially developed by Barros et al. (2000) and Barreiro et al. (2002).
Minor comments and edits:
- Figure 1. Replace “Wind (m s-1)” by “Streamlines” in the subtitles. Replace “wind (streamlines” by “streamlines” in Lines 51 and 52. Please describe the topography (brown shading) and explains SALLJ, SACZ, BH, and NL shown in Fig.1.
The mistakes in Figure 1 were fixed.
- Lines 55-56: What is “an annual cycle between September and April of the following year”?
The phrase “an annual cycle between September and April of the following year” was changed by “the wet season occurs between September and April of the following year” (lines 59-60).
- Figure 2 caption: Please explain “a.s.l.”
The abbreviation “a.s.l” was deleted.
- Line 88: Replace “describing climate dynamics” by “simulating climate changes.”
It was done (line 98).
- Lines 160-165: Please correct section numbers. For example, section 3 should be section 2
The sequence of the sections was fixed (lines 221-226).
- Line 256: What do new CMIP5 models mean?
The term “new” was deleted from the sentence (line 326).
- Lines 259-264: Should Fig. 2 here be Fig. 1?
The mistake was corrected.
- Lines 268-270: The statement is confusing. Positive OLR anomalies indicate suppressed convection.
The statement “The positive OLR anomalies indicate the dynamic link between convection over the Amazon basin and SACZ” was changed by “The positive OLR anomalies indicate that SACZ suppresses convection over the Amazon basin” (lines 343-344).
- Figure 3 caption: Please explain the value at the top right of panels b)-f). Also for Fig. 4.
These values represent the correlation coefficient between OLR spatial patterns from NCEP/NOAA and each GCM. The meaning of the numbers within parentheses was added in the captions of Figures 3 and 4.
- Line 287: It looks to me that there is no greater similarity .
We changed “have greater similarity with the” by “reproduce the observed OLR pattern” (line 363).
- Figure 5: Labels (a)-(h) are not consistent with those (a-i) described in the figure caption. The 200hPa wind field is not shown but streamlines. The unit (m/s) is also not correct.
The mistakes in the captions and titles of Figure 5 were fixed.
- Line 317: Are the results in Figure 5 from composite or just long-term mean climatology?
Figure 5 presents the long-term mean climatology for the 1980-2005 period.

Round 2
Reviewer 2 Report
Thanks for revising the paper as I suggested. I am happy with the revisions and I think the paper is ready for publication. The authors just forgot to delete lines 644 to 647.
Reviewer 4 Report
The authors have taken considerable steps in revising the manuscript, and comprehensively addressing my earlier comments. I recommend acceptance of the paper.